# Magnetic Resonance Imaging in the Neuroimaging of Progressive Supranuclear Palsy—Parkinsonism Predominant: Limitations and Strengths in Clinical Evaluation

**DOI:** 10.3390/diagnostics15080945

**Published:** 2025-04-08

**Authors:** Piotr Alster, Michał Kutyłowski, Natalia Madetko-Alster

**Affiliations:** 1Department of Neurology, Medical University of Warsaw, Kondratowicza 8, 03-242 Warsaw, Poland; natalia.madetko@wum.edu.pl; 2Department of Radiology, Mazovian Brodno Hospital, Kondratowicza 8, 03-242 Warsaw, Poland; michael.kutylowski@gmail.com

**Keywords:** atypical Parkinsonism, neuroimaging, PSP, Parkinsonian syndromes, MRI

## Abstract

Progressive Supranuclear Palsy (PSP) is an atypical Parkinsonism, pathologically described as a four-repeat tauopathy. The contemporary criteria for diagnosis of PSP indicate akinesia, oculomotor dysfunction, postural instability, and language/cognitive impairment as core symptoms. Among these features, the first two are linked to PSP—Parkinsonism predominant (PSP-P). PSP-P is the second most common subtype of PSP, following PSP—Richardson’s syndrome (PSP-RS), and is associated with a more gradual deterioration, beneficial course, and longer life expectancy after diagnosis. It is also problematic in terms of clinical evaluation, as this entity may overlap with Parkinson’s disease (PD) in early stages and with other atypical Parkinsonisms in more advanced stages. The evolution in understanding PSP and the possible progress in care and therapy of the disease leads to the necessity of finding optimal examination methods with sufficient sensitivity and specificity. In this context, PSP-P seems a crucial point. The goal of this narrative review is to provide an overview of the possibilities provided by Magnetic Resonance Imaging (MRI) assessments in terms of PSP-P and analyze their strengths and weaknesses.

## 1. Introduction

Progressive Supranuclear Palsy (PSP) is the most common atypical Parkinsonism. Neuropathologically, it is characterized by four-repeat tauopathy with coiled oligodendroglia and tufted astrocytes, which are interpreted as hallmarks of the disease [1]. Contemporarily, its definite diagnosis is based on neuropathological assessment, with in vivo examinations leading to probable or possible diagnoses of PSP [2]. The causes of the disease have not yet been fully recognized; however, among the factors that likely contribute to the pathogenesis of PSP, the disruption of mitophagy, oxidative stress, and neuroinflammation can be mentioned [3]. It is not recognized whether neuroinflammation in PSP is rather a cause or consequence of the disease. Recently, the significance of assessing inflammatory factors in the evolution of certain subtypes of PSP has been stressed [4]. Moreover, it has not yet been verified whether the role of these factors is favorable or not [5]. To date, the possible significance of the combined role of inflammation and environmental exposure has been discussed [6,7]. The disease may be impacted by certain risk factors, such as age, diabetes, and hypertension [8,9]. Similar features impacting the disease can be observed in Corticobasal Syndrome, which is a clinical manifestation of various pathologies including PSP [10]. Generally, PSP is considered a sporadic disease, and cases of familial inheritance are rarely observed [11]. The key genetic factor related to PSP is associated with *MAPT* dysregulation, which is linked to astroglial tau deposition [12]. The pathophysiological aspects of the disease have been described using tau patterns and atrophic changes, which provide an overview of its staging [13]. Among the crucial regions evaluated in staging, the ventral diencephalon, pallidum, brainstem, striatum, amygdala, thalamus, frontal lobe, and occipital lobe can be mentioned [14]. Based on the Movement Disorders Society criteria for diagnosis, PSP is classified as a group of clinically differing subtypes, among which PSP-RS and PSP-P are the most common [2]. The neuropathological patterns differ between the subtypes of the disease. In the case of PSP—Richardson’s syndrome (PSP-RS) the tau load is higher than in PSP—Parkinsonism predominant (PSP-P) [13]. In terms of the area of total tau load, pronounced differences can be observed between PSP with postural instability (PSP-PI) and PSP-P [13]. In the context of the boundaries between PSP-P and PSP-RS, more significant differences are observed in cell-type accumulation. Studies have shown that the discrepancies between the two entities are the most marked in the oligodendroglial and astroglial tau while being less conspicuous in the neuronal tau [13]. This possibly impacts the abnormalities, depending on the region of interest in PSP-P, and leads to differences in the comparisons to other phenotypes of the disease. PSP-P is linked with about 30–35% of PSP cases and is defined by oculomotor dysfunction and akinesia [2,15]. When compared to PSP-RS, PSP-P is associated with a less-relevant deterioration and preservation of response to levodopa treatment [16]. The cognitive deterioration in PSP-P is less marked [17]. In the early stages of the disease, PSP-P may be clinically indistinguishable from Parkinson’s disease (PD) [15]. Moreover, the disease duration in PSP-P is longer than in PSP-RS. In this context, there is growing interest in the search for factors that impact the evolution of the PSP subtype with a more beneficial prognosis; namely, PSP-P. This leads to the necessity to obtain optimal tools for the examination of PSP-P. Due to overlaps between clinical manifestations between PSP and other entities, the examination of this disease must be supported by supplementary methods. Among procedures that are possibly feasible for the evaluation and differential diagnosis in neuroimaging of PSP-P, Magnetic Resonance Imaging (MRI), single-photon emission computed tomography (SPECT), positron emission tomography (PET) and transcranial sonography (TCS) can be mentioned. The majority of studies have concentrated on the evaluations of MRI. This is related to the questionable utility of TCS, as the results do not differ from those associated with PD [18]. Meanwhile, although limited access and higher costs limit the feasibility of PET and SPECT, the examination provides more meticulous data. The goal of this review is to highlight and discuss the strengths and weaknesses of MRI examination regarding this clinical entity.

## 2. Magnetic Resonance Imaging (MRI)

The features associated with PSP which generally impact the manifestation in MRI are related to atrophic abnormalities, loss of neuromelanin, iron deposition, and microstructural damage [19]. Examination based on MRI in PSP-P was generally previously mainly performed during the differential diagnosis between PD and PSP-RS, while less data are based on the examination of discrepancies in its comparison with MSA-P. Evaluations of PSP-P have shown that the disease may not necessarily be linked with midbrain atrophy in the early stages [20]. The introduction of ratios associated with brainstem structures and MRI-derived parameters facilitated an examination of the entity (Figure 1). Most studies assessing PSP-P are associated with neuroimaging of the brainstem, as this region is considered to be less affected by the pathophysiological aspect of the disease than in the case of PSP-RS [16]. Evaluations of cell type-specific tau accumulation have indicated that the midbrain tegmentum and pontine base are affected by significant differences in oligodendroglial and neuronal tau between PSP-P and PSP-RS. Higher levels of tau accumulation were observed in PSP-RS, which could determine the progress of neurodegenerative and consequently, atrophic changes [13]. The significance of Pons/midbrain areas ratio in MRI measurements for the differentiation of PSP-P and PD is considered to be ambiguous and, in some cases, the efficiency of this method has been questioned [21,22]. The discrepancies in the outcomes are likely to be linked to the stage of PSP-P, as the initial lack of feasibility in using the parameter could lead to significant differences in the follow-up [22]. The differential diagnosis between PSP-P and PD showed lower values in length between the interpeduncular fossa and the center of the cerebral aqueduct at the mid-mammillary-body level (MTEGM) [22]. The midbrain area, midbrain to pons ratio, and pons to midbrain ratio have been indicated as efficiently discriminating factors in the comparison between PSP-P and PSP-RS [22,23]. Further analyses of the subtentorial parameters led to the Magnetic Resonance Parkinsonism Index (MRPI) [24], which is calculated using the following formula: pons area/midbrain area x middle cerebellar peduncle width/superior cerebellar peduncle width (Figure 2 and Figure 3) [25]. In PSP-P, the MRPI was indicated as a possibly efficient factor in differential diagnosis and predicting vertical supranuclear gaze palsy (VSGP) [26,27]. However, the MRPI has not been interpreted as a sufficient tool for the differentiation of PSP-P in all studies. Picillo et al., in an analysis of midbrain-based morphometric assessment based on the examination of 21 patients with PSP-P and 35 patients with PD, showed that the discriminatory features of the MRPI and pons-to-midbrain could not necessarily be optimal [28]. Moreover, higher diagnostic accuracy was observed in the pons-to-midbrain ratio when compared to MRPI. More efficient performance of the MRPI in the differentiation of PSP-P and PSP-RS was detected with the combined use of this parameter and volumetric/thickness data [29]. The inclusion of superior cerebellar peduncle width indicated that the indicated associations between this parameter and postural instability are possibly crucial for the differentiation of PSP-RS and PSP-P [30]. Superior cerebellar peduncle assessments of atrophic changes were also found to be more pronounced in PSP-RS when compared to PSP-P and PSP-speech/language [31]. In 2018, an evolution of this parameter was introduced, which was named MRPI 2.0. The factor is computed by multiplying the MRPI with the ratio of average width (from three measurements) of the third ventricle on an axial image at the level of anterior and posterior commissures and maximal left to right frontal horn width on the axial image in the anterior and posterior commissure plane (Figure 4 and Figure 5) [32]. The MRPI 2.0 provides higher accuracy than the MRPI for the differentiation of PSP-P patients from those with PD. The authors of the study on MRPI 2.0 stressed that, while both MRPI and MRPI 2.0 provide sufficient data to discriminate PSP-P with VSGP (O1), only MRPI 2.0 sufficiently discriminated PSP with the slowness of vertical saccades (O2) with PD [32]. On the other hand, the study by Picillo et al. indicated the possibly limited significance of MRPI 2.0 regarding the differentiation between PSP-P and PD [28]. The study did not provide additional sub-analyses on the evaluation of oculomotor dysfunction. The implementation of automated MRPI 2.0 values, instead of manual measurements, was indicated as an advance in the differentiation between early-stage PSP-P and PD [33]. Therefore, it can be assumed that the MRPI 2.0 should be considered as a useful parameter; however, specific PSP phenotyping should also be included in the evaluation.

Interestingly, another research work examining patients with PSP-P and MSA-P showed that, while the properties leading to differentiation are preserved in the case of mesencephalon to pons ratio and the MRPI, in the case of the MRPI 2.0 no significant differences between the diseases were observed [34]. Evaluations concerning the width of the third ventricle as a potentially feasible differential factor in the differentiation between PSP-P and MSA-P indicated more highlighted discrepancies between the two diseases in the examination of frontal perfusion in SPECT [35].

Evaluations of combined assessments of brain volume in MRI and ^18^F-flortaucipir indicated spared midbrain accompanied by increased accumulation of the radiotracer in the putamen among PSP-P patients [36]. Measurements regarding the tectal plate in PSP-RS, PSP-P, and PD revealed reduced values in both PSP groups when compared to PD and controls; however, atrophic changes were more pronounced in PSP-RS. The study indicated a correlation between the measured parameter and clinical deterioration in PSP [37]. Examinations of cortical abnormalities in PSP showed more pronounced atrophic changes in PSP-RS and partly spared cortex in PSP-P [23]. Verification of the laterality index in the context of cortical atrophy showed no difference between PSP subtypes [38]. Another study reported that evaluations bound to subcortical and cortical volumetric assessments do not provide sufficient data to differentiate PSP-P and PSP-RS; however, more pronounced abnormalities were observed in PSP-RS [29]. Additional atrophic changes in the central segment of the corpus callosum were detected in PSP-RS when compared to PSP-P [39]. In PSP-P, dominant atrophic changes are observed in the subcortical region [40].

There is growing interest in the assessment of PSP using Diffusion Tensor (DT) MRI. The research conducted by Agosta et al. [41] involved analyzing MRPI and DT in PSP-P and PSP-RS. It was found that infratentorial white matter and thalamic radiations were more impacted by the course of disease in PSP-RS when compared to PSP-P. The authors indicated that DT MRI analysis combined with MRPI evaluation provided increased discriminatory power. Evaluations using Voxel-Based Morphometry in PSP-P and PSP-RS demonstrated multiple differences between the subtypes. More pronounced grey matter loss was detected in the midbrain, left cerebellar lobe, and Date nuclei in PSP-RS [41]. Moreover, in the white matter, excessive atrophic changes were found within the midbrain, internal capsulae, and orbitofrontal, prefrontal, and precentral/premotor regions in a bilateral manner in PSP-RS, and in the frontal lobe bilaterally in PSP-P [41]. A comparison of PD, PSP-P, and PSP-RS based on DT MRI showed promising results in the evaluation of the dentatorubrothalamic tract [42]. In PSP-RS, the measurement deviated more from the norm. The study demonstrated the association of this factor with gait and postural dysfunction in the examined groups [42]. The significance of white matter abnormalities in PSP-P has been examined in a longitudinal study in which the evolution of destruction in the supratentorial tracts was interpreted as a useful factor in the in vivo examination of disease progression within this entity [43]. The same study indicated the relative preservation of the white matter in the cerebellum. Examinations of white matter changes in PSP-P, PSP-RS, and PD have also been performed using track-density imaging [44]. A support vector machine was implemented to differentiate the assessed groups. The outcome of the study showed that, in the context of track-density imaging, PSP-P was associated with deviations that were more pronounced than in PD and less than in PSP-RS. The abnormalities were detected within the midbrain, superior cerebellar peduncles, cerebellum, and corticospinal tract [44]. Studies on tractography in the differentiation of Parkinsonisms are still evolving, with the deviations in fiber tracts of PSP-RS being more stressed [45]. It was revealed that DTI imaging of the dentatorubrothalamic tract, the anterior thalamic radiation, and the volume of the dorsal part of the midbrain showed satisfactory discriminatory properties in the examination of PSP-RS, PSP-P, and PD. In all the considered parameters PSP-P was indicated as an intermediate form of neurodegeneration; however, the mentioned methods enabled differentiation of early-stage cases, which seems crucial in the context of obstacles to clinical evaluation. Other methods of MRI examination showed moderate discrimination between PSP-RS and PSP-P in terms of the superior frontooccipital in fractional anisotropy and superior cerebellar peduncle in mean diffusivity [31]. Brain iron content quantification using MRI within the red nucleus and substantia nigra indicated that no differences between PSP-P and PSP-RS can be observed [46]. However, sufficient differential properties—in the context of sensitivity and specificity—were detected in the comparison between PSP and control, without specification of the disease subtype. Preliminary studies on the significance of multi-voxel MRI spectroscopy in the examination of PSP subtypes showed differences in the levels of gamma-aminobutyric acid (GABA), glutamate, glutamine, tau, and the N-acetylaspartate (NAA)/creatinine ratio. It should be noted that the study was based on an assessment of very small groups of patients, including 12 patients with PSP-P and 4 with PSP—Cerebellar dominant [47]. Assessment of magnetic susceptibility in atypical Parkinsonisms and PD showed patterns related to specific entities [48]. In particular, the susceptibility within the pallidum, substantia nigra, red nucleus, and cerebellar dentate was greater in PSP-P and PSP-RS than among healthy volunteers. In the case of the red nucleus, significantly higher susceptibility was observed in PSP-P when compared to PD.

## 3. Discussion

MRI does not fully address the issue of PSP differentiation, as the discrepancies with respect to hydrocephalus are not necessarily visible [49,50]. The search for efficient MRI PSP-P evaluations should more extensively address the discrepancies in atrophic changes in the context of tau patterns. The observed abnormalities, which are a reflection of tau accumulation, do not provide a sufficient overview; this is partly proved by the frequently contradictory outcomes of the results, as is often the case in evaluations of the brainstem. While MRI could provide a basic perspective on the effect of distinction between total tau loading, the analysis of consequences of cell type variability in tau accumulation—which is currently evaluated—is not achievable with this neuroimaging methodology [13]. The fact that the abnormalities in PSP-P are associated with a different pattern of cell-specific tau accumulation than other PSP subtypes likely affects the specificity of neuroimaging methods, in a manner dependent on the stage of the disease. As a pathology, PSP is considered to evolve within the pallido-nigro-luysian system and spreads in the rostral direction, heading via the striatum and amygdala to the cerebral cortex and, eventually, to the brainstem [14]. The spread of the disease is described in PSP as a single entity; however, taking the abnormalities observed in certain phenotypes into account, it can be assumed that a wider analysis should be performed, depending on the phenotype of the disease.

The contemporary literature suggests that the examination of PSP patients without taking into consideration different subtypes may be confusing, as the discrepancies observed in neuroimaging likely highlight different courses of pathomechanisms. The abnormalities observed in PSP-P and PSP-RS in MRI have recently been associated with abnormalities in glial-derived neurotrophic factor (GDNF) levels [51]. A negative correlation was found between the level of GDNF in the CSF and the M/P ratio in PSP-RS. In the same subtype, positive correlations were observed with the MRPI and MRPI 2.0. In PSP-P, the GDNF in serum was found to be positively correlated with the measurements of the third ventricle and the MRPI 2.0 and negatively correlated with the M/P ratio and the middle cerebellar peduncle area [51]. An intriguing point of examination of PSP-P could be prospectively associated with the combined examination of neuroimaging and biochemical factors. At present, the majority of relevant studies are based on the examination of PSP as a single group, without indication of the disease’s subtypes. One of the studies evaluated serum levels of neurofilament light chain protein and T1-weighted measurements of third ventricle width/intracranial diameter ratio, which were found to be increased in PSP when compared to PD/controls [52]. Another study established links between the levels of interleukin and abnormalities within the brainstem [5]. The serum level of interleukin 1 was associated with a likely beneficial effect in the mesencephalon, whereas interleukin 6 in the serum was linked with atrophic changes in the superior cerebellar peduncle. None of the mentioned studies implemented a sub-analysis of PSP phenotypes; however, based on the discrepancies in the levels of inflammatory factors between PSP-P and PSP-RS in other studies, extended research in this area could be useful [4]. The patterns of tau accumulation indicated in the literature enable the possibly feasible evaluation of PSP through combined MRI and in vivo tau imaging.

The use of MRI in PSP-P could be considered as a tool in supplementary examination; however, the combined neuroimaging of MRI with PET and SPECT may provide a broader perspective on mechanisms of evolution of atypical Parkinsonisms as PSP-P. This could be obtained by searching for possibly relevant links between regions of increased tau accumulation in PET and abnormalities in MRI. A possibly interesting point could be additionally obtained by evaluating overlaps between MRI and radiotracers indicating microglial activation in PET. This could lead to interesting observations in the context of the recently discussed significance of inflammation in neurodegeneration [53,54].

## 4. Conclusions and Future Perspectives

Perspectives concerning the implementation of therapeutic strategies for PSP lead to the necessity of efficient methods for its examination (Table 1). Inevitably, the existing methods are affected by significant limitations. The pathomechanism of PSP-P commonly indicates its more beneficial course in the context of sparing brainstem structures; however, in the evaluation of PSP-P in later stages, PSP-P and PSP-RS generally cannot be discriminated based on an examination of midbrain atrophy [39,55]. This results in obstacles linked to follow-up evaluations of PSP-P brainstem-derived MRI parameters. The vast majority of relevant studies have not included neuropathological evaluations, excluding the possibility of a definite diagnosis. Furthermore, studies focused on the examination of PSP have generally relied on the assessment of small groups from a single center, thus reducing the statistical significance of these studies. Some parameters were evaluated manually, excluding the possibility of fully repeatable examinations. Moreover, some of the literature conducted analyses of subtypes only retrospectively, limiting the ability for further examination and interpretation [17]. Another relevant aspect of these studies, which complicates the interpretation of neuroimaging results, is the fact that the subtypes of PSP generally evolve to the manifestation of PSP-RS. In this context, the evolution of MRI techniques could provide a wider view of the stages in the pathophysiological course of PSP. The current range of options related to the MRI-based evaluation of PSP allows only for a general overview and may not provide sufficient distinction regarding the less-clear clinical manifestations. Based on the contemporary literature, no technique is available to differentiate early PSP-P from PD in a single patient. Specific phenotyping of PSP clinical manifestations including multiple (preferably automated) MRI parameters in the analysis seems to be crucial for studies concerning those features facilitating differential diagnosis. However—at least currently—due to the relatively low specificity, MRI remains only an additional examination modality favoring everyday clinical practice. As such, further research in the field is required.

## Figures and Tables

**Figure 1 diagnostics-15-00945-f001:**
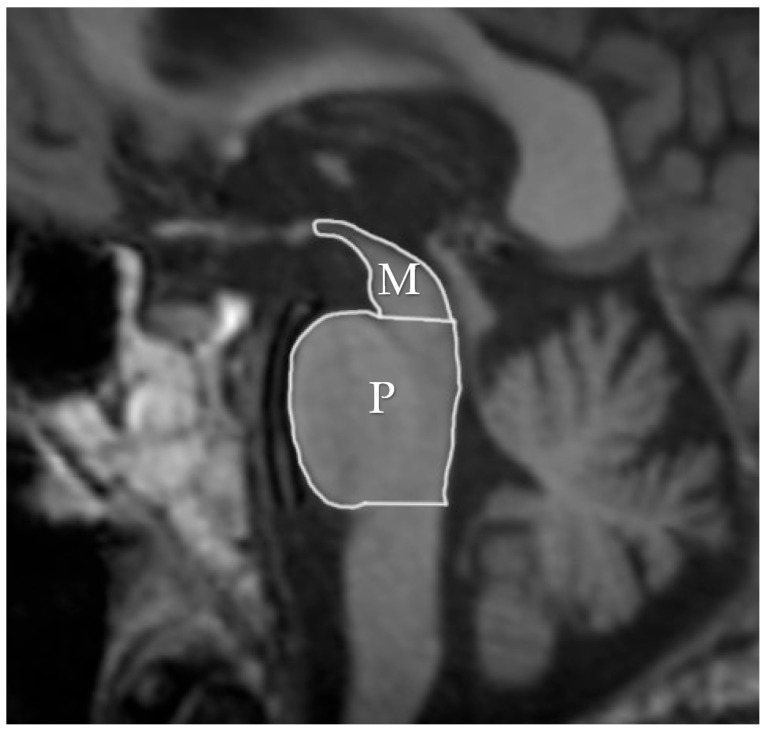
Areas of the midbrain (M) and pons (P).

**Figure 2 diagnostics-15-00945-f002:**
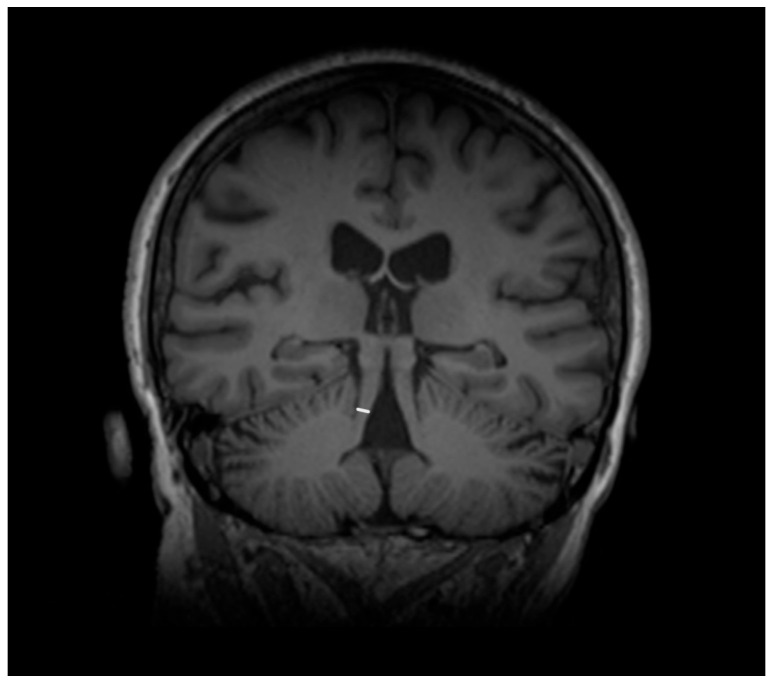
Width of the superior cerebellar peduncle.

**Figure 3 diagnostics-15-00945-f003:**
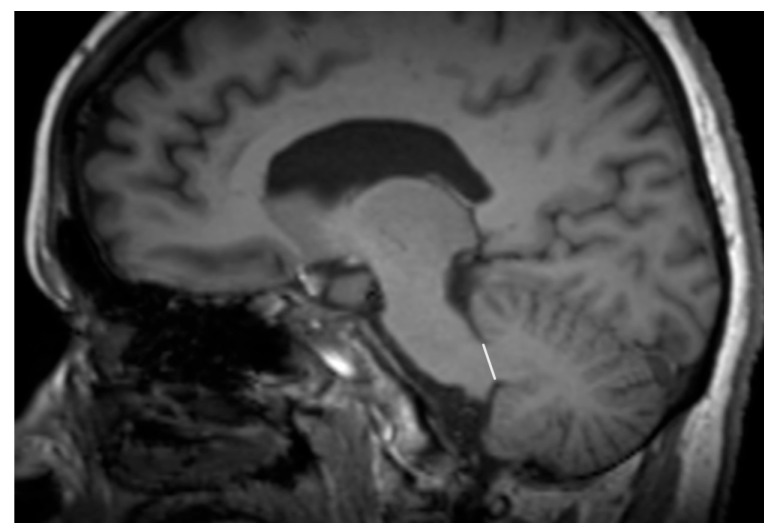
Width of the middle cerebellar peduncle.

**Figure 4 diagnostics-15-00945-f004:**
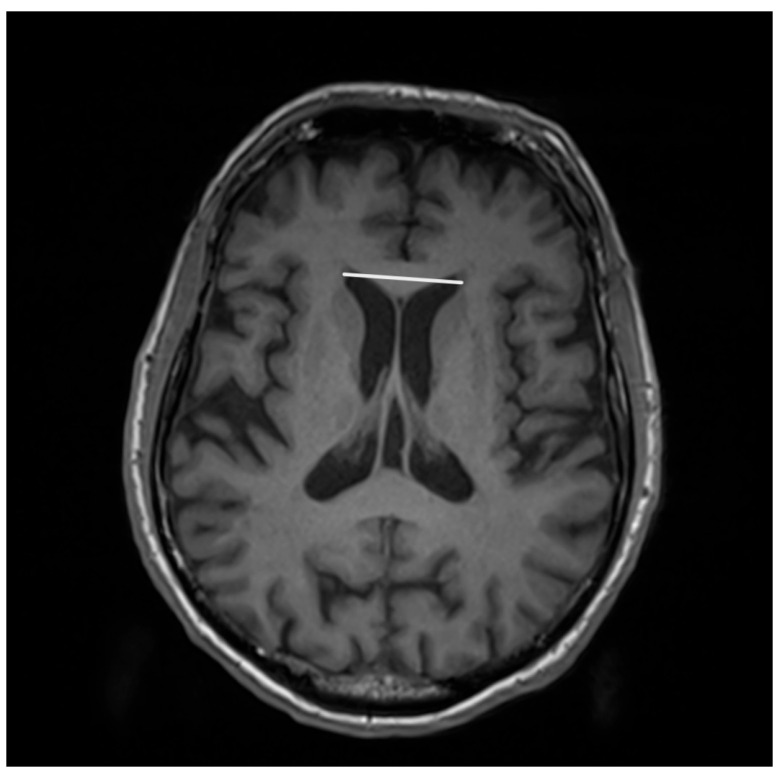
Maximal left to right frontal horn width on the axial image in the anterior and posterior commissure plane.

**Figure 5 diagnostics-15-00945-f005:**
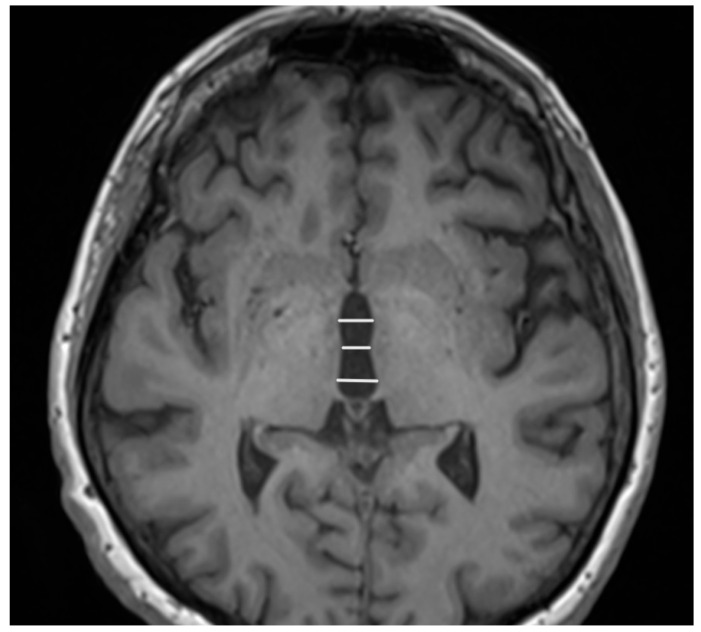
Width (from three measurements) of the third ventricle on an axial image at the level of anterior and posterior commissures.

**Table 1 diagnostics-15-00945-t001:** Significance of MRI in the differentiation of PSP-P and other entities.

Entity Differentiated with PSP-P	Significance in MRI
PD	- Midbrain to pons ratio - MRPI - MRPI 2.0 - DTI (dentatorubrothalamic tract) - Magnetic susceptibility
PSP-RS	- Midbrain to pons ratio - MRPI - MRPI 2.0 - Assessment of Superior Cerebellar Peduncle area - DTI (infratentorial white matter and thalamic radiations; midbrain; internal capsulae; orbitofrontal, prefrontal and precentral/premotor regions; frontal lobe dentatorubrothalamic tract) - Voxel-Based Morphometry - Fractional anisotropy (frontooccipital fasciculus) - Mean diffusivity (superior cerebellar peduncle) - Magnetic susceptibility
MSA-P	Midbrain to pons ratio MRPI
Other	Spectroscopy with PSP-C

DTI, Diffusion Tensor Imaging; MRI, Magnetic Resonance Imaging; MRPI, Magnetic Resonance Parkinsonism Index; MRPI 2.0, Magnetic Resonance Parkinsonism Index 2.0; MSA-P, Multiple System Atrophy—Parkinsonism Predominant; PSP-P, Progressive Supranuclear Palsy—Parkinsonism Predominant; PSP-RS, Progressive Supranuclear Palsy—Richardson’s Syndrome.

## Data Availability

No new data were created or analyzed in this study. Data sharing is not applicable to this article.

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
