# Peer review of "Magnetic Resonance Imaging in the Neuroimaging of Progressive Supranuclear Palsy—Parkinsonism Predominant: Limitations and Strengths in Clinical Evaluation"

_diagnostics, 2025, doi:10.3390/diagnostics15080945_

Round 1
Reviewer 1 Report
Comments and Suggestions for Authors
This is a potentially interesting narrative review on the use of MRI to differentiate PSP-P from other clinical entities. The revision of the literature is complete, but the layout of the manuscript is not easy to follow with some improper use of the language and several spelling errors.
Here are a few comments, to which the authors may respond in order to improve the clarity of the manuscript:
Page 1, last paragraph:
…The disease may be impacted by certain risk factors as age, diabetes and hypertension [8]. Concerning hypertension as a possible risk factor for PSP, this has been widely debated and for the sake of completeness I would also cite Colosimo C, Osaki Y, Vanacore N, Lees AJ. Mov Disord. 2003 Jun;18(6):694-7.
Page 2, line 53: better to write…Based on the MDS criteria of diagnosis, PSP is classified…
Same page, line 67: the current layout of the phrase is confusing. Please change to…In the early stages of the disease, PSP- P may clinically indistinguishable with Parkinson’s disease (PD) [14].
SAME page, line 121: Superior cerebellar peduncle assessments of atrophic changes were also found to be more stressed (?? ) in PSP-RS… Please clarify the meaning of stressed
Same page, line 151: I guess it is… Another paper
Page 4, line 169: …the measurement was more deviated? Please clarify the term deviated
Same page, line 181, change to…in fiber tracts
Page 5, line 204: the phrase…as its usefulness is possibly questioned in certain features as searching for discrepancies with other diseases, as idiopathic normal-pressure hydrocephalus…is convoluted and difficult to understand
Page 6, line 253: please change to …The vast majority of mentioned studies…
Table 1: fasciculus is misspelled
Finally, I think the author should mention in the conclusion that at the moment no MRI technique is available in a single patient to clearly distinguish early PSP-P from PD
Comments on the Quality of English LanguageTo be improved
Author Response
Dear Reviewer 1,
We would like to thank the Reviewers for all of their valuable suggestions and comments. We tried to approach all of the issues raised by the Reviewers. We hope that the given below explanations will be satisfactory. Accordingly, point to point responses can be found below. Authors highlighted in red the changed parts of the manuscripts.
Best regards
Piotr Alster
Page 1, last paragraph:
…The disease may be impacted by certain risk factors as age, diabetes and hypertension [8]. Concerning hypertension as a possible risk factor for PSP, this has been widely debated and for the sake of completeness I would also cite Colosimo C, Osaki Y, Vanacore N, Lees AJ. Mov Disord. 2003 Jun;18(6):694-7.
Authors are grateful for this comment. This change was implemented.
Page 2, line 53: better to write…Based on the MDS criteria of diagnosis, PSP is classified…
Authors are grateful for this comment. This change was implemented.
Same page, line 67: the current layout of the phrase is confusing. Please change to…In the early stages of the disease, PSP- P may clinically indistinguishable with Parkinson’s disease (PD) [14].
Authors are grateful for this comment. This change was implemented.
SAME page, line 121: Superior cerebellar peduncle assessments of atrophic changes were also found to be more stressed (?? ) in PSP-RS… Please clarify the meaning of stressed
Authors are grateful for this comment. This change was implemented.
Same page, line 151: I guess it is… Another paper
Authors are grateful for this comment. This change was implemented.
Page 4, line 169: …the measurement was more deviated? Please clarify the term deviated
Authors are grateful for this comment. This change was implemented.
Same page, line 181, change to…in fiber tracts
Authors are grateful for this comment. This change was implemented.
Page 5, line 204: the phrase…as its usefulness is possibly questioned in certain features as searching for discrepancies with other diseases, as idiopathic normal-pressure hydrocephalus…is convoluted and difficult to understand
Authors are grateful for this comment. This change was implemented.
Page 6, line 253: please change to …The vast majority of mentioned studies…
Authors are grateful for this comment. This change was implemented.
Table 1: fasciculus is misspelled
Authors are grateful for this comment. This change was implemented.
Finally, I think the author should mention in the conclusion that at the moment no MRI technique is available in a single patient to clearly distinguish early PSP-P from PD
Authors are grateful for this comment. This change was implemented.
In the attachment authors are providing the Language Editing Certificate.

Reviewer 2 Report
Comments and Suggestions for Authors
It is a review of the current literature on the evaluation of PSP-P with magnetic resonance imaging (MRI) and focuses on an important issue. However, due to some methodological and content deficiencies, the article was not deemed suitable for publication in its current form. Some corrections would be useful for further evaluation:
The article repeats information already available in the literature and does not offer a new perspective, an original hypothesis or a significant contribution. What is expected in review articles is not only the transfer of existing information but also its critical analysis and reaching new conclusions.
It does not clearly state which databases were searched in the study and which criteria were used to select or exclude studies. It contains a methodological weakness due to being a non-systematic review.
The current narrative in the article is superficial. While a more detailed discussion is expected to determine the sensitivity and specificity of MRI in the diagnosis of PSP-P, a more in-depth evaluation should be made on how MRI can better detect biomarkers associated with PSP-P.
While an analysis discussing the advantages and disadvantages of MRI evaluation of PSP-P compared to other neuroimaging methods (e.g. PET or SPECT) is expected, this issue has not been addressed sufficiently.
The language of the article is inconsistent at times and should be edited in a more professional way in terms of academic writing.
As a result, the article cannot be accepted in its current form. It is recommended that it be rewritten with a more systematic method, the role of MRI in PSP-P diagnosis be examined in more depth, and the implications for clinical practice be strengthened.
Comments on the Quality of English LanguageThe language of the article is inconsistent at times and should be edited in a more professional way in terms of academic writing.
Author Response
Dear Reviewer 2,
We would like to thank the Reviewers for all of their valuable suggestions and comments. We tried to approach all of the issues raised by the Reviewers. We hope that the given below explanations will be satisfactory. Accordingly, point to point responses can be found below. Authors highlighted in red the changed parts of the manuscripts.
Best regards
Piotr Alster
It is a review of the current literature on the evaluation of PSP-P with magnetic resonance imaging (MRI) and focuses on an important issue. However, due to some methodological and content deficiencies, the article was not deemed suitable for publication in its current form. Some corrections would be useful for further evaluation:The article repeats information already available in the literature and does not offer a new perspective, an original hypothesis or a significant contribution. What is expected in review articles is not only the transfer of existing information but also its critical analysis and reaching new conclusions.
The work was revised in the context of stressing strengths and weaknesses of the topic. Authors provided a revised review in a more critical manner.
It does not clearly state which databases were searched in the study and which criteria were used to select or exclude studies. It contains a methodological weakness due to being a non-systematic review.
The work is classified as an opinion, it is not a systematic review. Therefore authors intended to adjust the requirements indicated by the journal.
The current narrative in the article is superficial. While a more detailed discussion is expected to determine the sensitivity and specificity of MRI in the diagnosis of PSP-P, a more in-depth evaluation should be made on how MRI can better detect biomarkers associated with PSP-P.
The work was revised in the context of stressing strengths and weaknesses of the topic. Authors provided a revised review in a more critical manner – „Opinions are short articles that reflect the author’s viewpoints on a particular subject, technique, or recent findings. They should highlight the strengths and weaknesses of the topic presented in the opinion. The structure is similar to a review; however, they are significantly shorter and focused on the author’s view rather than a comprehensive, critical review.”
While an analysis discussing the advantages and disadvantages of MRI evaluation of PSP-P compared to other neuroimaging methods (e.g. PET or SPECT) is expected, this issue has not been addressed sufficiently.
Authors agree that the issue of PET and SPECT would be an interesting point of view, however it would be exceed the boundaries of the scope of this manuscript.
The language of the article is inconsistent at times and should be edited in a more professional way in terms of academic writing.
Authors would like to thank for this comment. The language of the work was revised.
As a result, the article cannot be accepted in its current form. It is recommended that it be rewritten with a more systematic method, the role of MRI in PSP-P diagnosis be examined in more depth, and the implications for clinical practice be strengthened.
The journal provides several options of types of articles. According to the journal’s website and the information from the journal, systematic review is not the only type of review published in this journal. The goal of this manuscirpt was to provide a narrative perspective on the issue of MRI in PSP-P. Systematic reviews would be a valuable point in the evaluation of certain imaging aspects of PSP-P, however in the case of this manuscript, it would not come up with the requirements concerning „Opinion” as a type of article (According to the definition on the journal’s website: „Opinions are short articles that reflect the author’s viewpoints on a particular subject, technique, or recent findings. They should highlight the strengths and weaknesses of the topic presented in the opinion. The structure is similar to a review; however, they are significantly shorter and focused on the author’s view rather than a comprehensive, critical review”). The work was revised in the context of stressing strengths weaknesses of the topic.
Comments on the Quality of English Language
The language of the article is inconsistent at times and should be edited in a more professional way in terms of academic writing.
The language of the study was revised. In the attachment authors are providing a certificate.

Reviewer 3 Report
Comments and Suggestions for Authors
his is paper dexrbing brain MRI techniques significance in PSP diagnosis .
The article is written clarly with clear aims and presentations.
I have just one but important comments - the authors should add some pictures from MRI PSP patients presenting disussed brain MRI parameters
Author Response
Dear Reviewer 3,
We would like to thank the Reviewers for all of their valuable suggestions and comments. We tried to approach all of the issues raised by the Reviewers. We hope that the given below explanations will be satisfactory. Accordingly, point to point responses can be found below. Authors highlighted in red the changed parts of the manuscripts.
Best regards
Piotr Alster
This is paper dexrbing brain MRI techniques significance in PSP diagnosis .
The article is written clarly with clear aims and presentations.
I have just one but important comments - the authors should add some pictures from MRI PSP patients presenting disussed brain MRI parameters
Authors are grateful for this comment, however due to the limitations related to the type of the manuscript „opinion”, authors performed a work which is more a viewpoint on the specific issue of MRI imaging in PSP-P. Authors agree that presenting MRIs with the parameters would be beneficial in future research works, however in this perspective authors attempted to obtain a feasible review in the analysis of neuroimaging of PSP, however without the necessity of obtaining additional bioethical approval, which would be necessary if using pictures of MRI.
Round 2
Reviewer 1 Report
Comments and Suggestions for Authors
I am satisfied with the authors' response
Author Response
Authors would like to thank the Reviewer for the valuable comments which improved our work.
Reviewer 2 Report
Comments and Suggestions for Authors
It seems that the authors did not take into account any of the comments other than updating the text language specified in the previous revision. I had stated in the previous revision that the article could not be accepted for publication in its current form. I suggest they reconsider otherwise I will regretfully reject it.
Author Response
Dear Reviewer 2,
Authors are grateful for the comment, however several issue should be highlighted.
1) The Reviewer indicated "While an analysis discussing the advantages and disadvantages of MRI evaluation of PSP-P compared to other neuroimaging methods (e.g. PET or SPECT) is expected, this issue has not been addressed sufficiently.". The topic of PET and SPECT is beyond the scope of this manuscript. Additionally the methods assess different features and cannot be efficiently comparable to MRI. Nevertheless to address the concern, authors implemented an additional short part of the manuscript highlighted in red at the end of section 3. This could highlight the possible significance of combined evaluations using MRI and PET/SPECT.
2) The type of article is "Opinion", not "Systematic Review". Therefore the features which are typically associated with systematic reviews do not refer to this paper. The Reviewer in round 1 stated "It does not clearly state which databases were searched in the study and which criteria were used to select or exclude studies. It contains a methodological weakness due to being a non-systematic review.", which does not come up with the type of article indicated in this study "opinion". The use of MRI in PSP-P is a bounded topic, due to the fact that PSP is a rare disease, moreover the manuscript refers to one of the subtypes of the disease. Additionally the analysis of PubMed database used in this study revealed around 50 works, most of which using different methods of examination in MRI. Moreover certain parameters as MRPI 2.0 are is some studies assessed manually, while in other authomatically. This combined with the fact of low number of studies accompanied by the small number of examined patients with PSP-P, would likely downscore the outcome of possible systematic reviews.
3) The Reviewer indicated that the study does not provide a perspective on the discussed issue. Authors after round 1 of the review additionally expanded the context of clinical strengths and limitations of MRI in PSP-P. The significance of PSP-P differentiation with other entities as PSP-RS and PD was commented. This issue seems the most crucial in the context of clinical significance of MRI in PSP-P.
4) Several comments of the Reviewer provide general comments e.g. "It seems that the authors did not take into account any of the comments other than updating the text language specified in the previous revision." The Reviewer did not refer to the fact that some of the suggestions would be contrary to the guidelines of "Opinion" in "Diagnostics" journal. The issue is only commented by the reviewer "It seems that the authors did not take into account any of the comments other than updating the text language specified in the previous revision." This puts authors in the position of either obeying the guidelines of the journal or implementing the comments of the Reviewer who clearly stated that the first option would results in intending to force rejection of the article, which may seem controversial.
Best regards
Corresponding Author
Reviewer 3 Report
Comments and Suggestions for Authors
The authors corrected article. I understand the difficulties to obtain the consent form for brain MRI, as well as review type o paper. I recommend to accept the paper in current form
Author Response

(The authors gave the same response as above.)

Round 3
Reviewer 2 Report
Comments and Suggestions for Authors
It is seen that the authors did not make any changes in the article except for the grammer correction during the revision, and that they refrained from making improvements in the second revision. In this case, I have to reject the article regretted.
Comments on the Quality of English LanguageAlthough the article has a large extent grammer change, there are still deficiencies of spelling.
Author Response
The language revision was performed by the Language Department of MDPI. Other concerns indicated by the Reviewer were addressed in the previous rounds of the review.